# Efficient Exact Gradient Update for training Deep Networks with Very Large Sparse Targets

**Pascal Vincent**[*]**, Alexandre de Brébisson, Xavier Bouthillier**
Département d'Informatique et de Recherche Opérationnelle
Université de Montréal, Montréal, Québec, CANADA
[*] and CIFAR

## Abstract

An important class of problems involves training deep neural networks with sparse prediction *targets* of very high dimension $D$. These occur naturally in e.g. neural language models or the learning of word-embeddings, often posed as predicting the probability of next words among a vocabulary of size $D$ (e.g. $200\,000$). Computing the equally large, but typically non-sparse $D$-dimensional output vector from a last hidden layer of reasonable dimension $d$ (e.g. $500$) incurs a prohibitive $O(Dd)$ computational cost *for each example*, as does updating the $D \times d$ output weight matrix and computing the gradient needed for backpropagation to previous layers. While efficient handling of large sparse network inputs is trivial, the case of large sparse *targets* is not, and has thus so far been sidestepped with approximate alternatives such as hierarchical softmax or sampling-based approximations during training. In this work we develop an original algorithmic approach which, for a family of loss functions that includes squared error and spherical softmax, can compute the *exact* loss, gradient update for the output weights, and gradient for backpropagation, all in $O(d^2)$ per example instead of $O(Dd)$, remarkably without ever computing the $D$-dimensional output. The proposed algorithm yields a speedup of $\frac{D}{4d}$, i.e. two orders of magnitude for typical sizes, for that critical part of the computations that often dominates the training time in this kind of network architecture.

## 1 Introduction

Many modern applications of neural networks have to deal with data represented, or representable, as very large sparse vectors. Such representations arise in natural language related tasks, where the dimension $D$ of that vector is typically (a multiple of) the size of the vocabulary, and also in the sparse user-item matrices of collaborative-filtering applications. It is trivial to handle very large sparse inputs to a neural network in a computationally efficient manner: the forward propagation and update to the input weight matrix after backpropagation are correspondingly sparse. By contrast, training with very large sparse prediction *targets* is problematic: even if the target is sparse, the computation of the equally large network output and the corresponding gradient update to the huge output weight matrix are *not sparse* and thus computationally prohibitive. This has been a practical problem ever since Bengio et al. [1] first proposed using a neural network for learning a language model, in which case the computed output vector represents the probability of the next word and is the size of the considered vocabulary, which is becoming increasingly large in modern applications [2]. Several approaches have been proposed to attempt to address this difficulty essentially by sidestepping it. They fall in two categories:

- *Sampling or selection based approximations* consider and compute only a tiny fraction of the output's dimensions sampled at random or heuristically chosen. The reconstruction sampling of Dauphin et al. [3], the efficient use of biased importance sampling in Jean et al. [4], the use of

Noise Contrastive Estimation [5] in Mnih and Kavukcuoglu [6] and Mikolov et al. [7] all fall under this category. As does the more recent use of approximate Maximum Inner Product Search based on Locality Sensitive Hashing techniques[8, 9] to select a good candidate subset.

- *Hierarchical softmax* [10, 7] imposes a heuristically defined hierarchical tree structure for the computation of the normalized probability of the target class.

Compared to the initial problem of considering all $D$ output dimensions, both kinds of approaches are crude approximations. In the present work, we will instead investigate a way to actually perform the *exact* gradient update that corresponds to considering *all* $D$ outputs, but do so implicitly, in a computationally efficient manner, without actually computing the $D$ outputs. This approach works for a relatively restricted class of loss functions, the simplest of which is linear output with squared error (a natural choice for sparse real-valued regression targets). The most common choice for multiclass classification, the *softmax* loss is not part of that family, but we may use an alternative *spherical softmax*, which will also yield normalized class probabilities. For simplicity and clarity, our presentation will focus on squared error and on an online setting. We will briefly discuss its extension to minibatches and to the class of possible loss functions in sections 3.5 and 3.6.

## 2   The problem

### 2.1   Problem definition and setup

We are concerned with gradient-descent based training of a deep feed-forward neural network with target vectors of very high dimension $D$ (e.g. $D = 200\,000$) but that are sparse, i.e. a comparatively small number, at most $K \ll D$, of the elements of the target vector are non-zero. Such a $K$-sparse vector will typically be stored and represented compactly as $2K$ numbers corresponding to pairs *(index, value)*. A network to be trained with such targets will naturally have an equally large output layer of dimension $D$. We can also optionally allow the input to the network to be a similarly high dimensional sparse vector of dimension $D_{in}$. Between the large sparse target, output, and (optionally large sparse) input, we suppose the network's intermediate hidden layers to be of smaller, more typically manageable, dimension $d \ll D$ (e.g. $d = 500$)[1].

**Mathematical notation:** Vectors are denoted using lower-case letters, e.g. $h$, and are considered column-vectors; corresponding row vectors are denoted with a transpose, e.g. $h^T$. Matrices are denoted using upper-case letters, e.g. $W$, with $W^T$ the transpose of $W$. The $i^{th}$ column of $W$ is denoted $W_i$ , and its $i^{th}$ row $W_{.i}$ (both viewed as a column vector). $U^{-T} = \left(U^{-1}\right)^T$ denotes the transpose of the inverse of a square matrix. $\mathbf{I}_d$ is the $d \times d$ identity matrix.

**Network architecture:** We consider a standard feed forward neural network architecture as depicted in Figure 1. An input vector $x \in \mathbb{R}^{D_{in}}$ is linearly transformed into a linear activation $a^{(1)} = W^{(1)T}x + b^{(1)}$ through a $D_{in} \times d$ input weight matrix $W^{(1)}$ (and an optional bias vector $b^{(1)} \in \mathbb{R}^d$). This is typically followed by a non-linear transformation $s$ to yield the representation of the first hidden layer $h^{(1)} = s(a^{(1)})$. This first hidden layer representation is then similarly transformed through a number of subsequent non-linear layers (that can be of any usual kind amenable to backpropagation) e.g. $h^{(k)} = s(a^{(k)})$ with $a^{(k)} = W^{(k)T}h^{(k-1)} + b^{(k)}$ until we obtain last hidden layer representation $h = h^{(m)}$. We then obtain the final $D$-dimensional network output as $o = Wh$ where $W$ is a $D \times d$ output weight matrix, which will be our main focus in this work. Finally, the network's $D$-dimensional output $o$ is compared to the $D$-dimensional target vector $y$ associated with input $x$ using squared error, yielding loss $L = \|o - y\|^2$.

**Training procedure:** This architecture is a typical (possibly deep) multi-layer feed forward neural network architecture with a *linear output layer* and *squared error loss*. Its parameters (weight matrices and bias vectors) will be trained by gradient descent, using gradient backpropagation [11, 12, 13] to efficiently compute the gradients. The procedure is shown in Figure 1. Given an example from the training set as an *(input,target)* pair $(x, y)$, a pass of forward propagation proceeds as outlined above, computing the hidden representation of each hidden layer in turn based on the previous one, and finally the network's predicted output $o$ and associated loss $L$. A pass of gradient backpropagation then works in the opposite direction, starting from $\nabla_o = \frac{\partial L}{\partial o} = 2(o - y)$ and

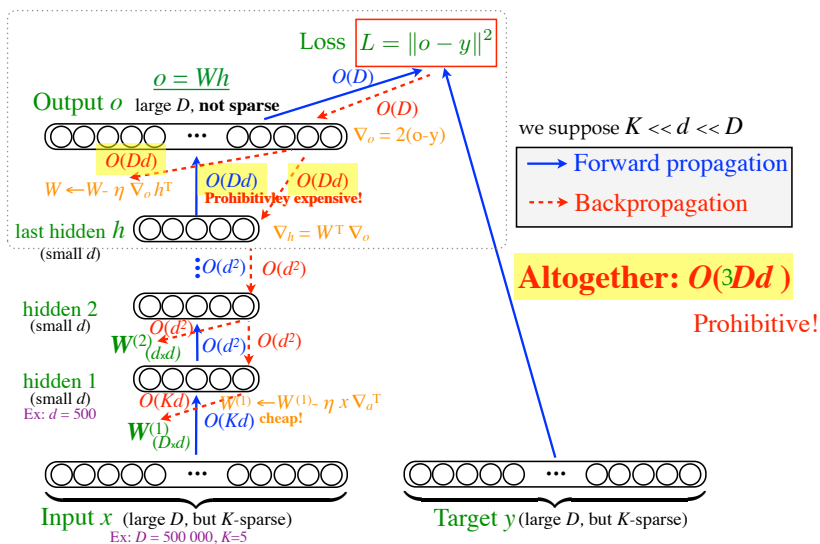

Figure 1: The computational problem posed by very large sparse targets. Dealing with sparse input efficiently is trivial, with both the forward and backward propagation phases easily achieved in $O(Kd)$. However this is not the case with large sparse targets. They incur a prohibitive computational cost of $O(Dd)$ at the output layer as forward propagation, gradient backpropagation and weight update each require accessing all $D \times d$ elements of the large output weight matrix.

propagating back the gradients $\nabla_{h^{(k)}} = \frac{\partial L}{\partial h^{(k)}}$ and $\nabla_{a^{(k)}} = \frac{\partial L}{\partial a^{(k)}}$ upstream through the network. The corresponding gradient contributions on parameters (weights and biases), collected along the way, are straightforward once we have the associated $\nabla_{a^{(k)}}$. Specifically they are $\nabla_{b^{(k)}} = \nabla_{a^{(k)}}$ and $\nabla_{W^{(k)}} = h^{(k-1)}(\nabla_{a^{(k)}})^T$. Similarly for the input layer $\nabla_{W^{(1)}} = x(\nabla_{a^{(1)}})^T$, and for the output layer $\nabla_W = (o - y)h^T$. Parameters are then updated through a gradient descent step $W^{(k)} \leftarrow W^{(k)} - \eta \nabla_{W^{(k)}}$ and $b^{(k)} \leftarrow b^{(k)} - \eta \nabla_{b^{(k)}}$, where $\eta$ is a positive learning-rate. Similarly for the output layer which will be our main focus here: $W \leftarrow W - \eta \nabla_W$.

## 2.2 The easy part: input layer forward propagation and weight update

It is easy and straightforward to efficiently compute the forward propagation, and the backpropagation and weight update part for the *input layer* when we have a very large $D_{in}$-dimensional but $K-$sparse input vector $x$ with appropriate sparse representation. Specifically we suppose that $x$ is represented as a pair of vectors $u, v$ of length (at most) $K$, where $u$ contains integer indexes and $v$ the associated real values of the elements of $x$ such that $x_i = 0$ if $i \notin u$, and $x_{u_k} = v_k$.

- **Forward propagation through the input layer:** The sparse representation of $x$ as the positions of $K$ elements together with their value makes it cheap to compute $W^{(1)T}x$. Even though $W^{(1)}$ may be a huge full $D_{in} \times d$ matrix, only $K$ of its rows (those corresponding to the non-zero entries of $x$) need to be visited and summed to compute $W^{(1)T}x$. Precisely, with our $(u, v)$ sparse representation of $x$ this operation can be written as $W^{(1)T}x = \sum_{k=1}^{K} v_k W^{(1)}_{:u_k}$ where each $W^{(1)}_{:u_k}$ is a $d$-dimensional vector, making this an $O(Kd)$ operation rather than $O(Dd)$.
- **Gradient and update through input layer:** Let us for now suppose that we were able to get gradients (through backpropagation) up to the first hidden layer activations $a^{(1)} \in \mathbb{R}^d$ in the form of gradient vector $\nabla_{a^{(1)}} = \frac{\partial L}{\partial a^{(1)}}$. The corresponding gradient-based update to input layer weights $W^{(1)}$ is simply $W^{(1)} \leftarrow W^{(1)} - \eta x(\nabla_{a^{(1)}})^T$. This is a rank-one update to $W^{(1)}$. Here again, we see that only the $K$ rows of $W^{(1)}$ associated to the (at most) $K$ non-zero entries of $x$ need to be modified. Precisely this operation can be written as: $W^{(1)}_{:u_k} \leftarrow W^{(1)}_{:u_k} - \eta v_k \nabla_{a^{(1)}} \quad \forall k \in \{1, \dots, K\}$ making this again a $O(Kd)$ operation rather than $O(Dd)$.

## 2.3 The hard part: output layer propagation and weight update

Given some network input $x$, we suppose we can compute without difficulty through forward propagation the associated last hidden layer representation $h \in \mathbb{R}^d$. From then on:

- Computing the final output $o = Wh$ incurs a prohibitive computational cost of $O(Dd)$ since $W$ is a full $D \times d$ matrix. Note that there is a-priori no reason for representation $h$ to be sparse (e.g. with a sigmoid non-linearity) but even if it was, this would not fundamentally change the problem since it is $D$ that is extremely large, and we supposed $d$ reasonably sized already. Computing the residual $(o - y)$ and associated squared error loss $\|o - y\|^2$ incurs an additional $O(D)$ cost.
- The gradient on $h$ that we need to backpropagate to lower layers is $\nabla_h = \frac{\partial L}{\partial h} = 2W^T(o - y)$ which is another $O(Dd)$ matrix-vector product.
- Finally, when performing the corresponding output weight update $W \leftarrow W - \eta(o-y)h^T$ we see that it is a rank-one update that updates all $D \times d$ elements of $W$, which again incurs a prohibitive $O(Dd)$ computational cost.

For very large $D$, all these three $O(Dd)$ operations are prohibitive, and the fact that $y$ is sparse, seen from this perspective, doesn't help, since neither $o$ nor $o - y$ will be sparse.

# 3 A computationally efficient algorithm for performing the exact online gradient update

Previously proposed workarounds are approximate or use stochastic sampling. We propose a different approach that results in the *exact same*, yet efficient gradient update, remarkably without ever having to compute large output $o$.

## 3.1 Computing the squared error loss $L$ and the gradient with respect to $h$ efficiently

Suppose that, we have, for a network input example $x$, computed the last hidden representation $h \in \mathbb{R}^d$ through forward propagation. The network's $D$ dimensional output $o = Wh$ is then in principle compared to the high dimensional target $y \in \mathbb{R}^D$. The corresponding squared error loss is $L = \|Wh - y\|^2$. As we saw in Section 2.3, computing it in the direct naive way would have a prohibitive computational complexity of $O(Dd + D) = O(Dd)$ because computing output $Wh$ with a full $D \times d$ matrix $W$ and a typically non-sparse $h$ is $O(Dd)$. Similarly, to backpropagate the gradient through the network, we need to compute the gradient of loss $L$ with respect to last hidden layer representation $h$. This is $\nabla_h = \frac{\partial L}{\partial h} = \frac{\partial \|Wh - y\|^2}{\partial h} = 2W^T(Wh - y)$. So again, if we were to compute it directly in this manner, the computational complexity would be a prohibitive $O(Dd)$. **Provided we have maintained an up-to-date matrix** $Q = W^TW$, which is of reasonable size $d \times d$ and can be cheaply maintained as we will see in Section 3.3, we can rewrite these two operations so as to perform them in $O(d^2)$:

**Loss computation:**

$$
\begin{aligned}
L &= \| \overbrace{Wh}^{O(Dd)} -y\|^2 \\
&= (Wh - y)^T (Wh - y) \\
&= h^T W^T W h - y^T W h - h^T W^T y + y^T y \\
&= h^T Q h - 2h^T(W^T y) + y^T y \\
&= h^T(\underbrace{Qh}_{O(d^2)} -2\underbrace{W^T y}_{O(Kd)}) + \underbrace{y^T y}_{O(K)} \qquad (1)
\end{aligned}
$$

**Gradient on $h$:**

$$
\begin{aligned}
\nabla_h = \frac{\partial L}{\partial h} &= \frac{\partial \|Wh - y\|^2}{\partial h} \\
&= 2W^T(Wh - y) \\
&= 2\left(W^T W h - W^T y\right) \\
&= 2(\underbrace{Qh}_{O(d^2)} - \underbrace{W^T y}_{O(Kd)}) \qquad (2)
\end{aligned}
$$

The terms in $O(Kd)$ and $O(K)$ are due to leveraging the $K$-sparse representation of target vector $y$. With $K \ll D$ and $d \ll D$, we get altogether a computational cost of $O(d^2)$ which can be several orders of magnitude cheaper than the prohibitive $O(Dd)$ of the direct approach.

### 3.2 Efficient gradient update of $W$

The gradient of the squared error loss with respect to output layer weight matrix $W$ is $\frac{\partial L}{\partial W} = \frac{\partial \|Wh - y\|^2}{\partial W} = 2(Wh - y)h^T$. And the corresponding gradient descent update to $W$ would be $W_{new} \leftarrow W - 2\eta(Wh - y)h^T$, where $\eta$ is a positive learning rate. Again, computed in this manner, this induces a prohibitive $O(Dd)$ computational complexity, both to compute output and residual $Wh - y$, and then to update all the $Dd$ elements of $W$ (since generally neither $Wh - y$ nor $h$ will be sparse). All $D \times d$ elements of $W$ must be accessed during this update. On the surface this seems hopeless. But we will now see how we can achieve the *exact* same update on $W$ in $O(d^2)$. The trick is to represent $W$ *implicitly* as the factorization $\underbrace{W}_{D \times d} = \underbrace{V}_{D \times d}\underbrace{U}_{d \times d}$ and update $U$ and $V$ instead

$$\text{a) } U_{new} = U - 2\eta(Uh)h^T \tag{3}$$
$$\text{b) } V_{new} = V + 2\eta y(U_{new}^{-T}h)^T \tag{4}$$

This results in *implicitly* updating $W$ as we did *explicitly* in the naive approach as we now prove:

$$
\begin{aligned}
V_{new}U_{new} &= (V + 2\eta y(U_{new}^{-T}h)^T)\,U_{new} \\
&= VU_{new} + 2\eta y(U_{new}^{-T}h)^T U_{new} \\
&= VU_{new} + 2\eta y h^T U_{new}^{-1} U_{new} \\
&= V(U - 2\eta(Uh)h^T) + 2\eta y h^T(U_{new}^{-1}U_{new}) \\
&= VU - 2\eta VUhh^T + 2\eta y h^T \\
&= VU - 2\eta(VUh - y)h^T \\
&= W - 2\eta(Wh - y)^T h^T \\
&= W_{new}
\end{aligned}
$$

We see that the update of $U$ in Eq. 3 is a simple $O(d^2)$ operation. Following this simple rank-one update to $U$, we can use the Sherman-Morrison formula to derive the corresponding rank-one update to $U^{-T}$ which will also be $O(d^2)$:

$$U_{new}^{-T} = U^{-T} + \frac{2\eta}{1 - 2\eta\|h\|^2}(U^{-T}h)h^T \tag{5}$$

It is then easy to compute the $U_{new}^{-T}h$, an $O(d^2)$ operation needed in Eq. 4. The ensuing rank-one update of $V$ in Eq 4, thanks to the $K$-sparsity of $y$ is only $O(Kd)$: only the $K$ rows $V$ associated to non-zero elements in $y$ are accessed and updated, instead of *all* $D$ rows of $W$ we had to modify in the naive update! Note that with the factored representation of $W$ as $VU$, we only have $W$ implicitly, so the $W^Ty$ terms that entered in the computation of $L$ and $\nabla_h$ in the previous paragraph need to be adapted slightly as $\hat{y} = W^Ty = U^T(V^Ty)$, which becomes $O(d^2 + Kd)$ rather than $O(Kd)$ in computational complexity. But this doesn't change the overall $O(d^2)$ complexity of these computations.

### 3.3 Bookkeeping: keeping an up-to-date $Q$ and $U^{-T}$

We have already seen, in Eq. 5, how we can cheaply maintain an up-to-date $U^{-T}$ following our update of $U$. Similarly, following our updates to $U$ and $V$, we need to keep an up-to-date $Q = W^TW$ which is needed to efficiently compute the loss $L$ (Eq. 1) and gradient $\nabla_h$ (Eq. 2). We have shown that updates to $U$ and $V$ in equations 3 and 4 are equivalent to implicitly updating $W$ as $W_{new} \leftarrow W - 2\eta(Wh - y)h^T$, and this translates into the following update to $Q = W^TW$:

$$
\begin{aligned}
\hat{z} &= Qh - U^T(V^Ty) \\
Q_{new} &= Q - 2\eta\left(h\hat{z}^T + \hat{z}h^T\right) + (4\eta^2 L)hh^T \tag{6}
\end{aligned}
$$

The proof is straightforward but due to space constraints we put it in supplementary material. One can see that this last bookkeeping operation also has a $O(d^2)$ computational complexity.

### 3.4 Putting it all together: detailed algorithm and expected benefits

We have seen that we can efficiently compute cost $L$, gradient with respect to $h$ (to be later back-propagated further) as well as updating $U$ and $V$ and performing the bookkeeping for $U^{-T}$ and $Q$. Algorithm 1 describes the detailed algorithmic steps that we put together from the equations derived above. Having $K \ll d \ll D$ we see that the proposed algorithm requires $O(d^2)$ operations, whereas the standard approach required $O(Dd)$ operations. If we take $K \approx d$, we may state more precisely that the proposed algorithm, for computing the loss and the gradient updates will require roughly $12d^2$ operations whereas the standard approach required roughly $3Dd$ operations. So overall the proposed algorithm change corresponds to a computational speedup by a factor of $\frac{D}{4d}$. For $D = 200\,000$ and $d = 500$ the expected speedup is thus **100**. Note that the advantage is not only in *computational* complexity, but also in *memory access*. For each example, the standard approach needs to access and change all $D \times d$ elements of matrix $W$, whereas the proposed approach only accesses the much smaller number $K \times d$ elements of $V$ as well as the three $d \times d$ matrices $U, U^{-T}$, and $Q$. So overall we have a **substantially faster algorithm**, which, while doing so *implicitly*, will nevertheless perform the *exact same* gradient update as the standard approach. We want to emphasize here that our approach is completely different from simply chaining 2 linear layers $U$ and $V$ and performing ordinary gradient descent updates on them: this would result in the same prohibitive computational complexity as the standard approach, and such ordinary separate gradient updates to $U$ and $V$ would not be equivalent to the ordinary gradient update to $W = VU$.

**Algorithm 1** Efficient computation of cost $L$, gradient on $h$, and update to parameters $U$ and $V$

| Step # | Operation | Computational complexity | Number of multiply-adds |
|---|---|---|---|
| 1: | $\hat{h} = Qh$ | $O(d^2)$ | $d^2$ |
| 2: | $\hat{y} = U^T(V^T y)$ | $O(Kd + d^2)$ | $Kd + d^2$ |
| 3: | $\hat{z} = \hat{h} - \hat{y}$ | $O(d)$ | $d$ |
| 4: | $\nabla_h = 2\hat{z}$ | $O(d)$ | $d$ |
| 5: | $L = h^T \hat{h} - 2h^T \hat{y} + y^T y$ | $O(2d + K)$ | $2d + K + 1$ |
| 6: | $U_{new} = U - 2\eta(Uh)h^T$ | $O(d^2)$ | $2d^2 + d$ |
| 7: | $U_{new}^{-T} = U^{-T} + \frac{2\eta}{1-2\eta\|h\|^2}(U^{-T}h)h^T$ | $O(d^2)$ | $2d^2 + 2d + 3$ |
| 8: | $V_{new} = V + 2\eta y(U_{new}^{-T}h)^T$ | $O(d^2 + Kd)$ | $d^2 + K + Kd$ |
| 9: | $Q_{new} = Q - 2\eta\left(h\hat{z}^T + \hat{z}h^T\right) + (4\eta^2 L)hh^T$ | $O(d^2)$ | $4 + 2d + 3d^2$ |
| | **Altogether:** | $O(d^2)$ provided $K < d \ll D$ | $\approx 12d^2$ elementary operations |

### 3.5 Controlling numerical stability and extension to the minibatch case

The update of $U$ in Equation 3 may over time lead $U$ to become ill-conditioned. To prevent this, we regularly (every 100 updates) monitor its conditioning number. If either the smallest or largest singular value moves outside an acceptable range[2], we bring it back to 1 by doing an appropriate rank-1 update to $V$ (which costs $Dd$ operations, but is only done rarely). Our algorithm can also be straightforwardly extended to the minibatch case (the derivations are given in the supplementary material section) and yields the same theoretical speedup factor with respect to the standard naive approach. But one needs to be careful in order to keep the computation of $U^{-T}h$ reasonably efficient: depending on the size of the minibatch $m$, it may be more efficient to solve the corresponding linear equation for each minibatch from scratch rather than updating $U^{-T}$ with the Woodbury equation (which generalizes the Sherman-Morrison formula for $m > 1$).

### 3.6 Generalization to a broader class of loss functions

The approach that we just detailed for linear output and squared error can be extended to a broader, though restricted, family of loss functions. We call it the *spherical family of loss functions* because it includes the spherical alternative to the softmax, thus named in [14]. Basically it contains any loss function that can be expressed as a function of only the $o_c$ associated to non-zero $y_c$ and of $\|o\|^2 = \sum_j o_j^2$ the squared norm of the whole output vector, which we can compute cheaply, irrespective of $D$, as we did above[3]. This family does not include the standard softmax loss $\log \frac{\exp(o_c)}{\sum_j \exp(o_j)}$, but it does include the *spherical softmax*[4]: $\log \frac{o_c^2 + \epsilon}{\sum_j (o_j^2 + \epsilon)}$. Due to space constraints we will not detail this extension here, only give a sketch of how it can be obtained. Deriving it may not appear obvious at first, but it is relatively straightforward once we realize that: a) the gain in computing the squared error loss comes from being able to very cheaply compute the sum of squared activations $\|o\|^2$ (a scalar quantity), and will thus apply equally well to other losses that can be expressed based on that quantity (like the spherical softmax). b) generalizing our gradient update trick to such losses follows naturally from gradient backpropagation: the gradient is first backpropagated from the final loss to the scalar sum of squared activations, and from there on follows the same path and update procedure as for the squared error loss.

## 4 Experimental validation

We implemented both a CPU version using *blas* and a parallel GPU (Cuda) version using *cublas* of the proposed algorithm[5]. We evaluated the GPU and CPU implementations by training word embeddings with simple neural language models, in which a probability map of the next word given its preceding n-gram is learned by a neural network. We used a Nvidia Titan Black GPU and a i7-4820K @ 3.70GHz CPU and ran experiments on the one billion word dataset[15], which is composed of 0.8 billions words belonging to a vocabulary of 0.8 millions words. We evaluated the resulting word embeddings with the recently introduced Simlex-999 score [16], which measures the similarity between words. We also compared our approach to unfactorised versions and to a two-layer hierarchical softmax. Figure 2 and 3 (left) illustrate the practical speedup of our approach for the output layer only. Figure 3 (right) shows that out LST (Large Sparse Target) models are much faster to train than the softmax models and converge to only slightly lower Simlex-999 scores. Table 1 summarizes the speedups for the different output layers we tried, both on CPU and GPU. We also empirically verified that our proposed factored algorithm learns the model weights $(VU)$ as the corresponding naive unfactored algorithm's $W$, as it theoretically should, and followed the same learning curves (as a function of number of iterations, not time!).

## 5 Conclusion and future work

We introduced a new algorithmic approach to efficiently compute the *exact* gradient updates for training deep networks with very large sparse targets. Remarkably the complexity of the algorithm is independent of the target size, which allows tackling very large problems. Our CPU and GPU implementations yield similar speedups to the theoretical one and can thus be used in practical applications, which could be explored in further work. In particular, neural language models seem good candidates. But it remains unclear how using a loss function other than the usual *softmax* might affect the quality of the resulting word embeddings so further research needs to be carried out in this direction. This includes empirically investigating natural extensions of the approach we described to other possible losses in the spherical family such as the *spherical-softmax*.

**Acknowledgements:** We wish to thank Yves Grandvalet for stimulating discussions, Çağlar Gülçehre for pointing us to [14], the developers of Theano [17, 18] and Blocks [19] for making these libraries available to build on, and NSERC and Ubisoft for their financial support.

Table 1: Speedups with respect to the baseline naive model on CPU, for a minibatch of 128 and the whole vocabulary of D = 793471 words. This is a model with two hidden layers of $d = 300$ neurons.

| Model | output layer only speedup | whole model speedup |
|---|---|---|
| cpu unfactorised (naive) | 1 | 1 |
| gpu unfactorised (naive) | 6.8 | 4.7 |
| gpu hierarchical softmax | 125.2 | 178.1 |
| cpu factorised | 763.3 | 501 |
| gpu factorised | 3257.3 | 1852.3 |

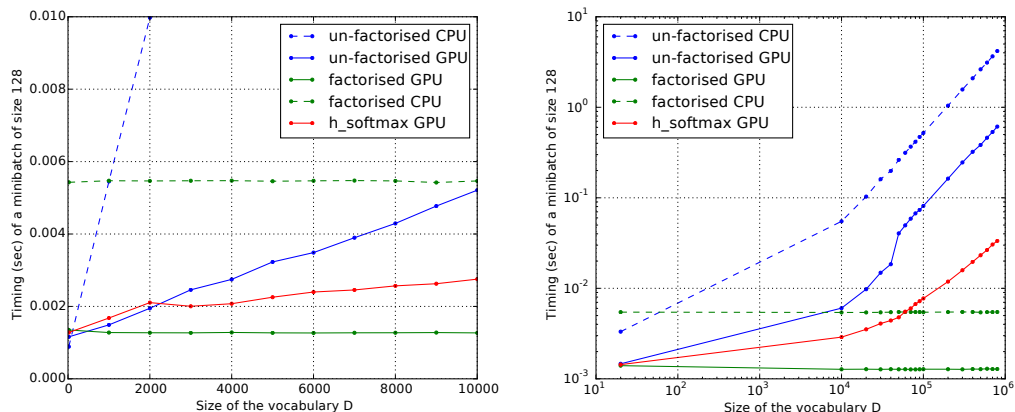

Figure 2: Timing of different algorithms. Time taken by forward and backward propagations in the output layer, including weight update, on a minibatch of size 128 for different sizes of vocabulary D on both CPU and GPU. The input size d is fixed to 300. The Timing of a 2 layer hierarchical softmax efficient GPU implementation (h_softmax) is also provided for comparison. Right plot is in log-log scale. As expected, the timings of factorized versions are independent of vocabulary size.

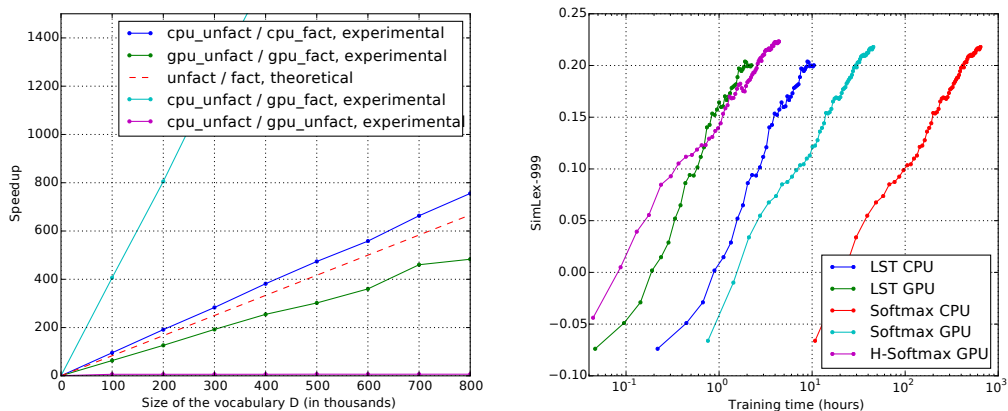

Figure 3: **Left:** Practical and theoretical speedups for different sizes of vocabulary D and fixed input size d=300. The practical unfact / fact speedup is similar to the theoretical one. **Right**: Evolution of the Simlex-999 score obtained with different models as a function of training time (CPU softmax times were extrapolated from fewer iterations). Softmax models are zero hidden-layer models, while our large sparse target (LST) models have two hidden layers. These were the best architectures retained in both cases (surprisingly the softmax models with hidden layers performed no better on this task). The extra non-linear layers in LST may help compensate for the lack of a softmax. LST models converge to slightly lower scores at similar speed as the hierarchical softmax model but significantly faster than softmax models.

## Footnotes

[1]Our approach does not impose any restriction on the architecture nor size of the hidden layers, as long as they are amenable to usual gradient backpropagation.

[2]More details on our numerical stabilization procedure can be found in the supplementary material

[3]In addition loss functions in this family are also allowed to depend on $\mathrm{sum}(o) = \sum_j o_j$ which we can also compute cheaply without computing $o$, by tracking $\bar{w} = \sum_j W_{:j}$ whereby $\mathrm{sum}(o) = \sum_j W_{:j}^T h = \bar{w}^T h$.

[4]where $c$ is the correct class label, and $\epsilon$ is a small positive constant that we added to the spherical interpretation in [14] for numerical stability: to guarantee we never divide by 0 nor take the log of 0.

[5]Open source code is available at: https://github.com/pascal20100/factored_output_layer

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
