[Supplementary Material]

# Efficient Exact Gradient Update for training Deep Networks with Very Large Sparse Targets
# [ Supplementary Material ]

**Pascal Vincent**
Département d'Informatique et de Recherche Opérationnelle
Université de Montréal
Montréal, Québec, CANADA
and CIFAR
vincentp@iro.umontreal.ca

**Alexandre de Brébisson**
Département d'Informatique et de Recherche Opérationnelle
Université de Montréal
Montréal, Québec, CANADA
alexandre.de.brebisson@umontreal.ca

**Xavier Bouthillier**
Département d'Informatique et de Recherche Opérationnelle
Université de Montréal
Montréal, Québec, CANADA
xavier.bouthillier@iumontreal.ca

# Appendix

## A  Minibatch version of the algorithm

The algorithm we derived for online gradient is relatively straightforward to extend to the case of minibatches containing $m$ examples, and will still yield the same theoretical speedup factor with respect to the standard naive approach. One may want to be careful in order to keep the computation of $U^{-T}h$ (or ore precisely $U^{-T}H$ in the minibatch case) reasonably efficient. In the minibatch version presented below, we update $U^{-T}$ based on the Woodbury equation (which generalizes the Sherman-Morrison formula for $m > 1$ and involves inverting an $m \times m$ matrix). But depending on the size of the minibatch $m$, it may become more efficient to solve the corresponding linear equations for each minibatch from scratch every time, rather than inverting that $m \times m$ matrix. In which case we won't need to maintain an $U^{-T}$ at all.

---
**Algorithm 1** Minibatch version of the update algorithm

---
**Initialization**

- we can initialize $D \times d$ matrix $V$ randomly as we would have initialized $W$ so that we initially have $V = W$.
  Alternatively we can initialize $V$ to 0 (there won't be symmetry breaking issues with having $W$ initially be 0 provided the other layers are initialized randomly, since varying inputs and targets will naturally break symmetry for the output layer)
- initialize $Q \leftarrow V^T V$ (or more cheaply initialize $Q \leftarrow 0$ if we have initialized $V$ to 0).
- we initialize $U$ to the identity: $U \leftarrow \mathbf{I}_d$ so that, trivially, we initially have $VU = W$.
- initialize $U^{-T} \leftarrow \mathbf{I}_d$

**Update**

We suppose we receive $m$ target vectors in the $m$ columns of sparse matrix $Y$, and corresponding $m$ hidden representations in the $m$ columns of matrix $H$.

| Step # | Operation | Computation complexity | Computational complexity with the multiplicative factor left in. |
|---|---|---|---|
| 1: | $\hat{H} = QH$ | $O(md^2)$ | $O(md^2)$ |
| 2: | $\hat{Y} = U^T(V^TY)$ | $O(mKd + md^2)$ | $O(mKd + md^2)$ |
| 3: | $\hat{Z} = \hat{H} - \hat{Y}$ | $O(md)$ | $O(md)$ |
| 4: | $\nabla_H = 2\hat{Z}$ | $O(md)$ | $O(md)$ |
| 5: | $M = H^T\hat{Z} - \hat{Y}^T H + Y^T Y$ or alternatively $M = H^T\hat{H} - (\hat{Y}^T H + H^T\hat{Y}) + Y^T Y$ | $O(m^2d + m^2K)$ | $O(2m^2d + m^2K)$ |
| 6: | $L = \text{Tr}(M)$ | $O(m)$ | $O(m)$ |
| 7: | $U_{new} = U - 2\eta(UH)H^T$ | $O(md^2)$ | $O(2md^2)$ |
| 8: | $U_{new}^{-T} = U^{-T} - (U^{-T}H)\left((H^TH - \frac{1}{2\eta}\mathbf{I}_m)^{-1}H^T\right)$ | $O(m^2d + m^3 + md^2)$ | $O(2m^2d + m^3 + 2md^2)$ |
| 9: | $V_{new} = V + 2\eta Y(U_{new}^{-T}H)^T$ | $O(md^2 + mKd)$ | $O(md^2 + mKd)$ |
| 10: | $Q_{new} = Q - 2\eta\left(H\hat{Z}^T + \hat{Z}H^T\right) + 4\eta^2(HM)H^T$ | $O(md^2 + dm^2)$ | $O(3md^2 + m^2d)$ |

---

# B    Detailed proof for computation of update of $Q$

Update to $Q$ corresponds to $W_{new} \leftarrow W - 2\eta(WH - Y)H^T$. We will use the following precomputed quantities: $Q = W^TW$, $\hat{H} = QH$ and $\hat{Y} = W^TY = U^T(V^TY)$ and $\hat{Z} = \hat{H} - \hat{Y}$.

$$
\begin{aligned}
Q_{new} &= W_{new}^T W_{new} \\
&= \left(W - 2\eta(WH - Y)H^T\right)^T \left(W - 2\eta(WH - Y)H^T\right) \\
&= W^TW - 2\eta H(WH - Y)^TW - 2\eta W^T(WH - Y)H^T \\
&\quad + 4\eta^2 H(WH - Y)^T(WH - Y)H^T \\
&= Q - 2\eta\left(HH^TW^TW - HY^TW\right) - 2\eta\left(W^TWHH^T - W^TYH^T\right) \\
&\quad + 4\eta^2 H(H^TW^TWH - H^TW^TY - Y^TWH + Y^TY)H^T \\
&= Q - 2\eta\left(HH^TQ - H(W^TY)^T\right) - 2\eta\left(QHH^T - (W^TY)H^T\right) \\
&\quad + 4\eta^2 H(H^TQH - H^T(W^TY) - (W^TY)^TH + Y^TY)H^T \\
&= Q - 2\eta\left(H\hat{H}^T - H\hat{Y}^T + \hat{H}H^T - \hat{Y}H^T\right) \\
&\quad + 4\eta^2 H(H^T\hat{H} - H^T\hat{Y} - \hat{Y}^TH + Y^TY)H^T \\
&= Q - 2\eta\left(H(\hat{H} - \hat{Y})^T + (\hat{H} - \hat{Y})H^T\right) + 4\eta^2 H(H^T(\hat{H} - \hat{Y}) - \hat{Y}^TH + Y^TY)H^T \\
&= Q - 2\eta\left(H\hat{Z}^T + \hat{Z}H^T\right) + 4\eta^2 H \underbrace{\left(H^T\hat{Z} - \hat{Y}^TH + Y^TY\right)}_{M} H^T
\end{aligned}
$$

This is what is listed as step 10 of the above minibatch algorithm.

In the online case, this becomes:

$$
\begin{aligned}
Q_{new} &= Q - 2\eta\left(h\hat{z}^T + \hat{z}h^T\right) + 4\eta^2\left(h^T\hat{z} - \hat{y}^Th + y^Ty\right)hh^T \\
&= Q - 2\eta\left(h\hat{z}^T + \hat{z}h^T\right) + 4\eta^2\left(h^T\hat{h} - h^T\hat{y} - \hat{y}^Th + y^Ty\right)hh^T \\
&= Q - 2\eta\left(h\hat{z}^T + \hat{z}h^T\right) + 4\eta^2\left(h^T\hat{h} - 2h^T\hat{y} + y^Ty\right)hh^T \\
&= Q - 2\eta\left(h\hat{z}^T + \hat{z}h^T\right) + (4\eta^2 L)hh^T
\end{aligned}
$$

which is the update listed as step 9 in the online algorithm.

## C  Details regarding controlling numerical stability

The update of $U$ (step 6 of the online algorithm, step 7 in the minibatch version) may over time lead to $U$ becoming ill-conditioned. Simultaneously, as we update $U$ and $U^{-T}$ (using Sherman-Morrison or Woodbury) our updated $U^{-T}$ may numerically start to diverge from the true $U^{-T}$ due to numerical precision. It is thus important to prevent both of these form happening, i.e. make sure $U$ stays well conditioned, to ensure the numerical stability of the algorithm. We present here progressively refined strategies for achieving this.

**Restoring the system in a pristine stable state**

One simple way to ensure numerical stability is to once in a while restore the system in its pristine state where $V = W$ and $U = \mathbf{I}_d = U^{-T}$. This is easily achieved as follows:

$$
\begin{aligned}
V &\leftarrow VU \\
U &\leftarrow \mathbf{I}_d \\
U^{-T} &\leftarrow \mathbf{I}_d.
\end{aligned}
$$

This operation doesn't affects the product $VU$, so the implicit matrix $W$ remains unchanged, nor does it affect $Q = W^T W$. And it does restore $U$ to a perfectly well conditioned identity matrix. But computing $VU$ is an extremely costly $O(Dd^2)$ operation, so if possible we want to avoid it (except maybe once at the very end of training, if we want to compute the actual $W$). In the next paragraphs we develop a more efficient strategy.

**Stabilizing only problematic singular values**

$U$ becoming ill-conditioned is due to its singular values over time becoming too large and/or too small. Let use define $\sigma_1$, ..., $\sigma_d$ as the singular values of $U$ ordered in decreasing order. The conditioning number of $U$ is defined as $\frac{\sigma_1}{\sigma_d}$ and it can become overly large when $\sigma_1$ becomes too large and/or when $\sigma_d$ becomes too small. Restoring the system in its pristine state, as shown in the previous paragraph, in effect brings back *all* singular values of $U$ back to 1 (since it brings back $U$ to being the identity). It is instead possible, and computationally far less costly, to correct when needed only for the singular values of $U$ that fall outside a safe range. Most often we will only need to occasionally correct for one singular value (usually the smallest, and only when it becomes too small). Once we have determined the offending singular value and its corresponding singular vectors, correcting for that singular value, i.e. effectively bringing it back to 1, will be a $O(Dd)$ operation. The point is to apply corrective steps only on the problematic singular values and only when needed, rather than blindly, needlessly and inefficiently correcting for all of them through the basic $O(Dd^2)$ full restoration explained in the previous paragraph.

The detailed algorithm that achieves this is given on the next page.

---
**Algorithm 2** Numerical stabilization procedure for problematic singular values
---
- The chosen safe range for singular values is $[\sigma_{\text{low}}, \sigma_{\text{high}}]$ (ex: $[0.001, 100]$ )
- The procedures given below act on output layer parameters $U, U^{-T}$ and $V$.
- For concision, we do not enlist these parameters explicitly in their parameter list.
- Procedure SINGULAR-STABILIZE gets called after every $n_{\text{check}}$ gradient updates (ex: $n_{\text{check}} = 100$).

**procedure** SINGULAR-STABILIZE( )
$\quad \bar{\mathbf{U}}, \sigma, \bar{\mathbf{V}} = \text{SVD}(U) \quad \triangleright$ Computes singular value decomposition of $U$ as $U = \bar{\mathbf{U}} \operatorname{diag}(\sigma) \bar{\mathbf{V}}^T$
$\quad$ **for all** $k \in \{1, \ldots, d\}$ **do**
$\quad\quad$ **if** $\sigma_k < \sigma_{\text{low}}$ OR $\sigma_k > \sigma_{\text{high}}$ **then**
$\quad\quad\quad$ FIX-SINGULAR-VALUE$(\sigma_k, \bar{\mathbf{U}}_k, 1)$
$\quad\quad$ **end if**
$\quad$ **end for**
**end procedure**

*The following procedure will change singular value $\sigma$ of $U$ associated to singular vector $u$ to become target singular value $\sigma^*$ (typically 1). It doesn't change $U$'s singular vectors, only that one singular value. It also changes $V$ symetrically (with a rank-one update) in such a way that $W = VU$ remains unchanged.*

**procedure** FIX-SINGULAR-VALUE$(\sigma, u, \sigma^*)$
$\quad \alpha = \frac{\sigma^* - \sigma}{\sigma}$
$\quad \beta = -\frac{\alpha}{1+\alpha}$
$\quad U \leftarrow U + \alpha u (U^T u)^T$
$\quad V \leftarrow V + \beta (Vu) u^T$
$\quad U^{-T} \leftarrow U^{-T} + \beta u (U^{-1} u)^T \qquad\qquad \triangleright$ Where $U^{-1}$ is obtained as the transpose of $U^{-T}$. But we may instead of this prefer to recompute $U^{-T}$ from scratch by inverting $U$ to ensure it doesn't stray too much due to numerical imprecisions.
**end procedure**
---

The proof that the FIX-SINGULAR-VALUE procedure achieves what it is supposed to is relatively straightforward, and left to the reader.

**Avoiding the cost of a full singular-value decomposition**

Computing the SVD of $d \times d$ matrix $U$ as required above, costs roughly $25d^3$ elementary operations (use the so-called R-SVD algorithm). But since the offending singular values will typically be only the smallest or the largest, it is wasteful to compute all $d$ singular values every time. A possibly cheaper alternative is to use the power iteration method with $U$ to find its largest singular value and associated singular vector, and similarly with $U^{-1}$ to obtain the smallest singular value of $U$ (which corresponds to the inverse of the largest singular value of $U^{-1}$). Each iteration of the power iteration method requires only $O(d^2)$ operations, and a few iterations may suffice. In our experiments we fixed it to 100 power iterations. Also it is probably not critical if the power iteration method is not run fully to convergence, as correcting along an approximate offending singular vector direction can be sufficient for the purpose of ensuring numerical stability. With this refinement, we loop over finding the smallest singular value with the power iteration method, correcting for it to be 1 by calling FIX-SINGULAR-VALUE if it is too small, and we repeat this until we find the now smallest singular value to be inside the acceptable range. Similarly for the largest singular values.

Note that while in principle we may not need to ever invert $U$ from scratch (as we provided update formulas of $U^{-T}$ with every change we make to $U$), it nevertheless proved to be necessary to do so regularly to ensure $U^{-T}$ doesn't stray too much from the correct value due to numerical imprecisions. Inverting $U$ using Gaussian-elimination costs roughly $d^3$ operations, so it is very reasonable and won't affect the computational complexity if we do it no more often than every $d$ training examples (which will typically correspond to less than 10 minibatches of size 128). In practice, we recompute $U^{-T}$ from scratch every time before we run this check for singular value stabilization.