[Reviews · NeurIPS 2015]

Submitted by Assigned_Reviewer_1

This paper develops an efficient algorithm for training neural networks to predict large sparse targets, such as words in a neural language model. In particular, it works for loss functions that only require access to the non-zero entries in the output and the squared norm of the predicted output vector. This excludes the traditional softmax layer, but spherical softmax can be used instead. Section 3 explains the factorization of the final layer matrix W and derives efficient computation in O(d^2) for the loss, the gradient w.r.t. the top hidden layer, and the gradient w.r.t. W. Section 4 validates the method in terms of mini-batch processing speed, relative speedup factor, and SimLex-999 scores. The proposed method is much faster than the naive unfactored versions, similar in speed to hierarchical softmax, and achieves slightly worse score compared to the hierarchical softmax.

I have included detailed comments below: - Section 2.3 : it looks like there is a change of notation of the target vector from "t" to "y". Are they indeed meant to be the same? - The abstract says that the proposed method works for "a family of loss functions" but the derivation only focuses on squared error. The family is not described until near the end of the paper, nor is it adequately explained why the proposed method works for precisely that family of loss functions. - It looks like there is a typo in equation 4; is there an extra lower-case u at the end? - L259: sited -> instead ? - Section 3.6 : This generalization beyond squared error is not proved anywhere. It is not obvious, at least not to me, why the claimed generalization is true.

Quality: The proposed method is very insightful, appears mathematically sound, and works well in practice.

Clarity: The writing is unfortunately a bit sloppy and the exposition could be clearer, especially regarding the type of loss functions for which this approach will work.

Originality: To my knowledge this work is quite original.

Significance: The work is potentially quite significant, especially if it can be extended beyond language modeling. Currently, it performs slightly worse than hierarchical softmax, so the near-term impact is somewhat limited. However, it is theoretically very appealing and I expect it could be useful for other problems with large, sparse targets.

Summary: The proposed layer factorization achieves impressive speedup with a modest tradeoff in the form of a restricted class of loss functions. I found this paper very clever and appreciated the empirical validation of the proposed method, although the presentation could be improved.

Submitted by Assigned_Reviewer_2

Summary:

This paper gives a novel technique for gradient-based learning in neural networks with a large but sparse target vector. The technique calculates the exact, correct gradient direction, while never computing the full large-dimensional output, albeit for a restricted class of loss functions. The paper shows, theoretically and empirically, that this drastically improves the computational complexity of gradient descent learning. Experiments document the speedup, but show that the loss function restrictions may negatively impact performance.

Major comments:

This paper presents a promising technique for speeding up neural language models with large, sparse output vectors. Remarkably, the paper shows how to perform exact gradient updates (for a restricted set of loss functions) in an efficient way that scales in the sparsity of the targets, rather than in the size of the output vector. This paper is not a tweak of previous approaches to dealing with large sparse targets, but a wholly new direction which may inspire a variety of related algorithms and improvements.

It will be no surprise to the authors that the paper would be strengthened greatly if it used the proposed method to attain state of the art performance on some task. The experiments in the paper show that the method indeed attains in practice the speedup promised, but this speedup is not leveraged to improve performance. It may be that the loss function restrictions are severe enough to stop this method from being practically useful. Most concerning is the comparison between hierarchical softmax and the proposed method. While H-softmax is only a heuristic, it is about equally fast and performs better than the proposed LST models on the example dataset in the paper. However the theoretical contributions of the paper are already substantial, and even if it takes some time to find a compelling application, the insights given here seem likely to be useful in some future application.

The experimental results are also valuable in showing that there are no numerical stability issues (other than the ill-conditioning of U which is identified and addressed).

The paper is very clearly written, with the algorithm spelled out step-by-step in useful tables. Additionally, code will be released upon publication, which should also aid clarity and adoption of the method in practice. The supplementary material adds the practically important detail of how to perform mini batch updates. The clarity of the algorithm could be further improved by explicitly including pseudocode for the rank-1 V update used to control numerical stability.

Minor comments:

pg 2 ln 097 "workFinally" ->"work. Finally" Fig 1 caption: "inO(Kd)"->"in O(Kd)" pg 4 ln 172: "non-lienarity"->"non-linearity" pg 5 ln 258: "theK"->"the K" pg 5 ln 259: "sited" -> "instead" pg 5 ln 261: "theW^Ty"->"the W^Ty" pg 7 ln 329: "appropriate do a rank-1" -> "appropriate rank-1" pg 7 ln 363: I don't believe the LST acronym has been introduced yet at this point in the text. pg 8 fig 2: the right hand plot does not appear to contain all points in the lefthand plot (the red curve never goes above the blue curve, for instance). Citation formats are inconsistent (some have whole name; some abbreviate first names; some say NIPS, some say Advances in Neural...) Citation 4 has a typo
Summary: This paper shows how to perform exact gradient updates in deep neural networks for sparse target vectors efficiently (for a restricted class of loss functions). This is an important contribution that may be widely built upon in the future, particularly in NLP applications, although the performance numbers reported in the paper are not yet state of the art.

Submitted by Assigned_Reviewer_3

The paper provides an efficient approach for dealing with output layers with large sparse targets. The algorithm is exact and not based on any approximations. The only downside of the approach is that it only applies to a limited number of loss functions (squared error and spherical softmax) and it is unclear whether the results can be extended to a much larger class of functions. Nevertheless, the approach will be useful for practical applications. Some minor issues:

- p.2 "workFinally", add period, space - p.4, "non-lienarity", should be "non-linearity" - p.5, Equation (4) has a spurious u-hat at the end - p.5, "sited of all ...", "instead of all ..." - p.5, "only theK rows", space missing? - p.5, "so theW^Ty", space missing? - p.6, "will however perform", "will nevertheless perform" - p.7, "Sheman-Morrison formula", add "r": "Sherman" - p.9, Entry [4] "Sebastien"
Summary: The paper is well written and presents a useful approach for reducing the computational complexity of neural network training (with restrictions on the loss function) when then output layer is large but sparse.

Author Feedback
Author rebuttal: We wish to thank the reviewers for carefully reading our paper, and for their thoughtful feedback that will help us improve it.

The following answers apply to comments and questions shared by several reviewers:

Regarding the generalization beyond squared error, we had planned to devote a full section to explain it in more details and had started writing the paper (abstract, intro, plan) accordingly. But due to space and time constraints the lengthier and less pedagogical derivation that generalizes our approach to a broader family of loss function didn't make it into the paper, and we were left with only a few lines mentioning it as a possibility. We are currently preparing an extended journal version that will explain and formally derive this generalization in full details, as well as provide a detailed pseudocode of our numerical stabilization procedure. For the NIPS conference paper, we propose to regroup the few lines that mention that the approach can be generalized to a broader loss function family towards the conclusion, briefly clarify what characterizes that family, and say that the detailed derivation of the extension will be addressed in future work. We will adjust the abstract accordingly.

Rest however assured that the same approach does indeed apply to a broader, though restricted, loss function family, that includes spherical softmax (we have also since then identified other interesting loss functions in that family). While it may not seem obvious at first how to generalize the approach, it becomes relatively straightforward once we realize that: a) the gain in computing the squared error loss comes from being able to very cheaply compute the sum of squared activations (a scalar quantity), and will thus apply equally well to other losses that can be expressed based on that quantity (like spherical softmax). b) generalizing our gradient update trick to such losses follows naturally from gradient backpropagation: the gradient is backpropagated from the final loss to the scalar sum of squared activations, and from there on follows the same path and update procedure as for the squared error loss.

We would of course love to showcase our method reaching state of the art performance on a compelling application (language-related tasks are the obvious first choice, but there are also other possibilities). We are actively working towards that goal. But we view it as outside of the scope of this paper, and linked to an orthogonal question that is largely open for future work: what are the merits and limitations of loss functions in the considered family, and what types of problems are they best suited for?
The focus of the current paper was on clearly explaining our original approach for computational speedup, and providing accurate empirical measurements of the actual speedups achieved.

We further thank the reviewers who did "heavy" reviews for catching typos and minor inconsistencies in the main text and references. (To reviewer 4: indeed we decided to change our notation for targets from t to y at some point, and forgot to change a couple of them.) These will all be fixed in the final paper.